

# Prevalence and risk factors of COVID-19-related generalized anxiety disorder among the general public in China: a cross-sectional study

Yi Xia[1],[*], Qi Wang[2],[*], Lushaobo Shi[1], Zengping Shi[1], Jinghui Chang[1], Richard Xu[3],[4], Huazhang Miao[1] and Dong Wang[1],[5],[6],[7]

[1] School of Health Management, Southern Medical University, Guangzhou, Guangdong, China
[2] Shenzhen People's Hospital, The First Affiliated Hospital, Southern University of Science and Technology, Shenzhen, Guangdong, China
[3] Department of Rehabilitation Sciences, The Hong Kong Polytechnic University, Hong Kong, China
[4] Center for Health Systems and Policy Research, Jockey Club School of Public Health and Primary Care, The Chinese University of Hong Kong, Hong Kong, China
[5] Institute of Health Management, Southern Medical University, Guangzhou, Guangdong, China
[6] Public Health Service System Construction Research Foundation of Guangzhou, Guangzhou, Guangdong, China
[7] Public Health Policy Research and Evaluation Key Laboratory Project of the Philosophy and Social Sciences of Guangdong College, Guangzhou, Guangdong, China
[*] These authors contributed equally to this work.

Corresponding author
Dong Wang, dongw96@smu.edu.cn

## ABSTRACT

**Objective:** This study aimed to estimate the prevalence of generalized anxiety disorder in China during the coronavirus disease 2019 (COVID-19) pandemic and identify its associated factors.

**Methods:** A cross-sectional study was conducted among the general population in China from March 16 to April 2, 2020. The participants were recruited using stratified random sampling. Data on demographic characteristics and COVID-19 related factors were obtained using self-administered questionnaires. The anxiety score was measured based on the Chinese version of the Generalized Anxiety Disorder 7-item Scale (GAD-7).

**Results:** The study comprised 10,824 participants, of which 37.69% had symptoms of anxiety. The risk factors for anxiety symptoms included poor self-reported health ($OR = 1.672$, $p < 0.001$), chronic diseases ($OR = 1.389$, $p < 0.001$), and quarantine ($OR = 1.365$, $p < 0.001$), while participants' perceptions that COVID-19 would be controlled was a protective factor ($OR = 0.774$, $p < 0.001$). The interactions between quarantine and self-reported health ($p < 0.001$), as well as between perceptions of COVID-19 and self-reported health ($p < 0.001$) were found to have a significant effect on GAD-7 scores.

**Conclusions:** Self-reported health status, chronic diseases, quarantine, and perceptions of COVID-19 were significantly associated with GAD-7 scores, indicating that mental health interventions are urgently needed during pandemics, especially for high-risk groups.

Subjects Psychiatry and Psychology, Public Health, Mental Health, COVID-19
Keywords COVID-19, Anxiety disorders, Public health, Epidemiology, Risk management

# INTRODUCTION

In December 2019, the coronavirus disease 2019 (COVID-19) occurred and rapidly spread throughout the world (*Sohrabi et al., 2020*). On January 31, 2020, the World Health Organization declared the outbreak a Public Health Emergency of International Concern and defined it as a pandemic on March 11, 2020 (*Vannabouathong et al., 2020*). Currently, COVID-19 has overwhelmed the health care system and imposed a heavy social and economic burden on individuals, families, communities, and countries (*Ferguson et al., 2020*; *Han et al., 2020*; *Bambra et al., 2020*).

The increasing number of COVID-19 cases may cause generalized anxiety disorder, which typically refer to a chronic mental disorder characterized by persistent, excessive, and irrational fear and tension. First, the high morbidity and mortality rates caused by the pandemic may lead to anxiety (*Mirhosseini et al., 2020*). Second, drastic lockdown measures, including quarantine, home isolation, restrictions on gatherings, and prohibition of many human activities may lead to unemployment and economic downturn, subsequently leading to anxiety (*Nussbaumer-Streit et al., 2020*; *Torales et al., 2020*). Finally, media coverage of diseases around the world may significantly influence or amplify individuals' anxiety levels (*Amirkhan, 2021*; *Li et al., 2020*). Excessive anxiety is harmful to the physical and mental health of the general public (*Kosic et al., 2020*). In addition to weakening the human immune system and increasing the risk of viral infections, it may also lead to panic, which is detrimental to the prevention and control of diseases (*Zhong et al., 2021*; *Fu et al., 2021*).

Recently, some studies have focused on assessing the degree of anxiety disorders in specific populations. For example, *Zhou et al. (2020)* reported that 45.4% of front-line medical staff suffered from anxiety. In addition, *Mosheva et al. (2020)* found that worries about infection, increasing workload, and increased sense of isolation placed medical staff under tremendous mental stress and led to high levels of anxiety. Furthermore, the prevalence of anxiety has been investigated among other groups, and its prevalence among college students, pregnant women, and teachers, was 41.1%, 21.7%, and 13.67%, respectively (*Fu et al., 2021*; *Shangguan et al., 2021*; *Li et al., 2020*). Although these studies have increased our understanding of anxiety during COVID-19, research relating to the general population is limited. Hence, this study assesses the prevalence of generalized anxiety disorder and its associated factors among the general population during the COVID-19 outbreak in China.

# MATERIALS AND METHODS

## Study design and participants

A cross-sectional web-based survey was conducted in China from March 16 to April 2, 2020. It adopted a combination of stratified random sampling and convenience sampling. First, according to China's geographical and economic development level, it was divided into three sub-regions (eastern, central, and western China)

(*National Bureau of Statistics, 2021*). Second, the Questionnaire Star electronic questionnaire platform (website: https://www.wjx.cn/) was used to complete the terminal output of the questionnaire, that was thereafter distributed to more than 30 regions in eastern, central, and western China through WeChat, QQ sharing, and web links. The sample size was expanded by snowball sampling during the survey. Participation was voluntary and anonymous. Inclusion criteria were: (a) respondents who consented to participate in the study after being informed of the survey's purpose and procedure, (b) respondents having access to the Internet *via* a computer or smartphone, and (c) respondents who had stayed in China during the COVID-19 pandemic. To reduce selection bias, the exclusion criteria were: (a) not completing the questionnaire and missing data, (b) answering time of more than 20 min or less than 2 min, and (c) the questionnaire having logic errors (*e.g.*, answers to all items of the questionnaire were the same). The target sample size was calculated according to the recommendation of 5–10 observed values per studied variable (*Jiang, 1997*). In this study, there were 14 variables, and the expected sample size was 140 (14 × 10). The study was conducted with the approval of the Ethics Committee of Southern Medical University (Ref ID: NFYKDX002).

## Patient and public involvement
The participants and the general public were not involved in the design, recruitment, and conduction of this study.

## Calculation of anxiety scores
The Chinese version of the Generalized Anxiety Disorder Scale-7 (GAD-7)—a valid self-reported psychometric scale—was used to measure individuals' anxiety levels in the past 2 weeks during COVID-19. The questionnaire comprised seven items using a four-point Likert scale ranging from 0 (not at all) to 3 (nearly every day), with a total score between 0–21 (*Ruiz et al., 2011*; *Johnson et al., 2019*). The levels of anxiety were classified as follows: normal (0–5), mild (5–9), moderate (10–14), and severe (15–21) (*Kroenke et al., 2007*). In a previous study (*Liu et al., 2020*), a total score higher than five was deemed as anxiety. This instrument has been widely used in China and also been confirmed to have good retesting reliability and validity (*Nyongesa et al., 2020*; *Zhang et al., 2021*). Similarly, in this study, the reliability was found to be very good (Cronbach's alpha = 0.94).

## Covariates for analysis
Socio-demographic variables in the analysis included sex, age, marital status, family registration, educational attainment, occupational background, and annual family income. In addition, the following health status data were collected and included: (1) good self-reported health (yes/no) and (2) chronic diseases (yes/no). Other factors included COVID-19-related variables: (1) Have you been diagnosed with COVID-19? (yes/no), (2) Have your family or friends been diagnosed with COVID-19? (yes/no), (3) Have you been in quarantine owing to COVID-19? (yes/no), and (4) What are your perceptions of COVID-19 (uncontrolled/controlled).

First, univariate analysis was performed to analyze the effects of socio-demographics and COVID-19-related factors on anxiety. Variables associated with the anxiety score through univariate analysis were included for subsequent multiple linear regression analysis. Furthermore, ordinal logistic regression analysis was used to assess the hierarchical relationship of the results.

## Ordinal logistic regression

Ordinal logistic regression was conducted to investigate the relationship between the risk factors and severity of anxiety, which was classified on the basis of participants' GAD-7 score outcomes: (1) A (minimal level of anxiety if the scores were <5), (2) B (mild level of anxiety if the scores were ≥5 but <10), and (3) C (moderate and severe level of anxiety if the scores were ≥10). To test the robustness of the results, the ordinal logistic regression model was further adjusted for demographics, including details relating to age, sex, and family, but as these did not change the results, only the analysis results, without adjusting for these variables, have been listed below.

## Interaction effects

The interaction was evaluated as a test of the coefficient of the product term formed by the factors involved. In this study, two interaction effects were assessed: (1) quarantine and self-reported health, and (2) perceptions of COVID-19 and self-reported health.

To understand the interaction effect between quarantine and self-reported health, the respondents were categorized into four groups: (a) quarantined respondents with poor self-reported health, (b) quarantined respondents with good self-reported health, (c) non-quarantined respondents with poor self-reported health, and (d) non-quarantined respondents with good self-reported health. The GAD-7 scores of these groups were then compared.

To analyze the interaction between perceptions of COVID-19 and self-reported health, the respondents were divided into four categories: (a) those who perceived that COVID-19 cannot be controlled with poor self-reported health, (b) those who perceived that COVID-19 cannot be controlled with good self-reported health, (c) those who perceived that COVID-19 can be controlled with poor self-reported health, and (d) those who perceived that COVID-19 can be controlled with good self-reported health. Thereafter, the GAD-7 score values of these groups were compared.

## Statistical analysis

In the present study, categorical variables were summarized as frequencies/percentages and continuous variables as means ± standard deviations (SD). Chi-squared tests were performed to compare categorical data. Univariate analysis was used to extract influence factors and multivariate regression analysis to examine the independent predictors of anxiety. To determine interaction effects, multiple group comparisons were conducted, and significant differences between the groups were evaluated by the Least Significant Difference test assuming equal variances or Tamhane's T2 (M) test assuming unequal

**Table 1 Associations between anxiety symptoms (GAD-7 score) with demographic characteristics.**

| Variable | Categories | Total (N = 10,824) | Anxiety symptoms (GAD-7 score) | | $\chi^2$ | p |
|---|---|---|---|---|---|---|
| | | | Yes (≥5) | No (<5) | | |
| Gender | Female | 5,827 (53.8%) | 2,304 (39.5%) | 3,523 (60.5%) | 18.315 | <0.001 |
| | Male | 4,997 (46.2%) | 1,776 (35.5%) | 3,221 (64.5%) | | |
| Age | 39 or below | 7,666 (70.8%) | 3,024 (39.4%) | 4,642 (60.6%) | 34.376 | <0.001 |
| | 40 or above | 3,158 (29.2%) | 1,056 (33.4%) | 2,102 (66.6%) | | |
| Marital status | Unmarried | 2,724 (25.2%) | 1,070 (39.3%) | 1,654 (60.7%) | 3.901 | 0.048 |
| | Married | 8,100 (74.8%) | 3,010 (37.2%) | 5,090 (62.8%) | | |
| Family registration | Rural | 2,724 (25.2%) | 1,129 (41.4%) | 1,595 (58.6%) | 21.823 | <0.001 |
| | Urban | 8,100 (74.8%) | 2,951 (36.4%) | 5,149 (63.6%) | | |
| Educational level | Secondary or below | 3,770 (34.8%) | 1,425 (37.8%) | 2,345 (62.2%) | 0.027 | 0.870 |
| | Tertiary or above | 7,054 (65.2%) | 2,655 (37.6%) | 4,399 (62.4%) | | |
| Occupational background | Medical background | 3,725 (34.4%) | 1,371 (36.8%) | 2,354 (63.2%) | 1.910 | 0.167 |
| | Non-medical background | 7,099 (65.6%) | 2,709 (38.2%) | 4,390 (61.8%) | | |
| Family annual income per year (RMB) | 100,000 or below | 7,374 (68.1%) | 2,827 (38.3%) | 4,547 (61.7%) | 4.078 | 0.043 |
| | 100,000 or above | 3,450 (31.9%) | 1,253 (36.3%) | 2,197 (63.7%) | | |

**Note:**
GAD-7, Generalized Anxiety Disorder-Scale-7; $p < 0.05$, statistically significant.

variances. A two-tailed $p < 0.05$ was considered as statistically significant. Statistical analysis was done with SPSS for Windows, version 25.0 (SPSS Inc., Chicago, IL, USA).

## RESULTS

### Participants' characteristics

A total of 10,980 respondents participated in the survey, but only 10,824 are included in the analysis, after excluding those who do not meet the inclusion criteria. The variation in the number of respondents is depicted through a flow chart (Fig. S1). The response rate to the questionnaire is 98.58%. The number of people in each province have been listed in Table S1. Among the 10,824 respondents, with a cutoff score of five, the overall prevalence of generalized anxiety disorder is 37.69%. Socio-demographic variables are presented in Table 1. Among the respondents, 53.83% are female ($p < 0.001$), 70.82% are aged <40 years ($p < 0.001$), 74.8% are married ($p = 0.048$), 74.8% come from urban areas ($p < 0.001$), and 68.1% have an annual family income less than RMB 100,000 ($p = 0.043$). Health status and COVID-19-related variables are presented in Table 2. While 17.3% respondents have poor self-reported health ($p < 0.001$), 5.64% have chronic diseases ($p < 0.001$), 10.6% have been quarantined ($p < 0.001$), and 88.8% perceive that COVID-19 can be controlled ($p < 0.001$). $p < 0.05$ indicate that GAD-7 scores are statistically different among these groups.

### Multiple regression analysis of factors associated with anxiety during COVID-19

Multiple linear regression analysis reveal that sex, age, family register, self-reported health, chronic diseases, quarantine, and perceptions of COVID-19 are significantly correlated

**Table 2 Associations between anxiety symptoms (GAD-7 score) with health status and COVID-19-related variables.**

| Variable | Categories | Total (N = 10,824) | Anxiety symptoms (GAD-7 score) | | $\chi^2$ | $p$ |
|---|---|---|---|---|---|---|
| | | | Yes (≥5) | No (<5) | | |
| Self-reported health | Poor | 1,867 (17.3%) | 903 (48.4%) | 964 (51.6%) | 109.418 | <0.001 |
| | Good | 8,957 (82.7%) | 3,177 (35.5%) | 5,780 (64.5%) | | |
| Chronic diseases | Yes | 611 (5.6%) | 278 (45.5%) | 333 (54.5%) | 16.797 | <0.001 |
| | No | 10,213 (94.4%) | 3,802 (37.2%) | 6,411 (62.8%) | | |
| Confirmed COVID-19 | Yes | 117 (1.1%) | 42 (35.9%) | 75 (64.1%) | 0.163 | 0.687 |
| | No | 10,707 (98.9%) | 4,038 (37.7%) | 6,669 (62.3%) | | |
| Relatives or friends confirmed | Yes | 112 (1.0%) | 48 (42.9%) | 64 (57.1%) | 1.285 | 0.257 |
| | No | 10,712 (99.0%) | 4,032 (37.6%) | 6,680 (62.4%) | | |
| Quarantine | Yes | 1,146 (10.6%) | 516 (45.0%) | 630 (55.0%) | 29.339 | <0.001 |
| | No | 9,678 (89.4%) | 3,564 (36.8%) | 6,114 (63.2%) | | |
| Perception of COVID-19 | Uncontrolled | 1,213 (11.2%) | 538 (44.4%) | 675 (55.6%) | 25.791 | <0.001 |
| | Controlled | 9,611 (88.8%) | 3,542 (36.9%) | 6,069 (63.1%) | | |

**Note:**
GAD-7, Generalized Anxiety Disorder-Scale-7; COVID-19, Coronavirus Disease 2019; $p < 0.05$, statistically significant.

**Table 3 Linear regression analysis of variables related to anxiety.**

| Variable | B | SE | Beta | t | 95% CI | r[*] | p[**] |
|---|---|---|---|---|---|---|---|
| Gender | 0.476 | 0.076 | 0.060 | 6.256 | [0.327–0.625] | 0.059 | <0.001 |
| Age | 0.581 | 0.090 | 0.066 | 6.440 | [0.404–0.758] | 0.061 | <0.001 |
| Marital status | 0.024 | 0.093 | 0.003 | 0.260 | [−0.159 to 0.207] | 0.002 | 0.795 |
| Family registration | 0.373 | 0.091 | 0.041 | 4.115 | [0.196–0.551] | 0.039 | <0.001 |
| Family annual income | −0.079 | 0.083 | −0.009 | −0.961 | [−0.242 to 0.083] | −0.009 | 0.337 |
| Self-reported health | 1.121 | 0.102 | 0.107 | 10.977 | [0.921–1.321] | 0.104 | <0.001 |
| Chronic diseases | 0.715 | 0.169 | 0.042 | 4.224 | [0.383–1.047] | 0.040 | <0.001 |
| Quarantine | 0.796 | 0.123 | 0.062 | 6.496 | [0.556–1.037] | 0.061 | <0.001 |
| Perception of COVID-19 | −0.963 | 0.12 | −0.076 | −8.062 | [−1.198 to −0.729] | −0.076 | <0.001 |

**Notes:**
CI, Confidence interval.
[*] Semi-partial correlations.
[**] $p$ values were calculated from linear regression models.

with generalized anxiety disorder (Table 3). Specifically, respondents who are female, are aged <40 years, and lived in rural areas are more likely to suffer from anxiety. In terms of health status, self-reported unhealthy participants and those with chronic diseases have higher levels of anxiety. Regarding COVID-19-related variables, quarantine is a risk factor for anxiety, while perceiving that COVID-19 can be controlled is a significantly protective factor for anxiety. Additionally, eastern, central, and western China data are separately analyzed (Tables S2 and S3). Furthermore, sensitivity analysis, namely logistic regression analysis, further confirm these results (Table S4).

## Association between risk factors with the severity of anxiety

For this analysis, participants are categorized as A, B, and C, as described in the Materials and Methods section. The number of participants with minimal (GAD-7 score <5), mild

**Table 4 Comparison of GAD-7 score in groups by different levels of quarantine and self-reported health and different levels of perception of COVID-19 and self-reported health.**

| Variable | Quarantine | | Row p (across quarantine) | Perception of COVID-19 | | Row p (across controlled) |
|---|---|---|---|---|---|---|
| | Quarantined | Non-quarantined | | Uncontrolled | Controlled | |
| Poor self-reported health | 214 (2.0%) | 1,653 (15.3%) | 0.002 | 276 (2.5%) | 1,591 (14.7%) | <0.001 |
| | 5.43 ± 4.695 | 4.46 ± 4.332 | | 5.75 ± 5.414 | 4.37 ± 4.148 | |
| Good self-reported health | 932 (8.6%) | 8,025 (74.1%) | <0.001 | 937 (8.7%) | 8,020 (74.1%) | <0.001 |
| | 4.09 ± 4.271 | 3.27 ± 3.789 | | 4.18 ± 4.760 | 3.26 ± 3.718 | |
| Column p (across self-reported health) | <0.001 | <0.001 | <0.001 | <0.001 | <0.001 | <0.001 |

Note:
Generalized Anxiety Disorder-Scale-7; $p < 0.05$, statistically significant.

(GAD-7 score ≥5 but <10), and moderate and severe anxiety (GAD-7 score ≥10) are 6,744, 3,355, and 725, respectively. Ordinal logistic regression analysis reveal that self-reported health, chronic diseases, quarantine, and perceptions of COVID-19 are all significantly associated with the severity of anxiety (Table S5).

**Interaction effect for GAD-7 scores**

Three different analyses are undertaken to evaluate the main effects of quarantine and self-reported health, as well as their interactions with the GAD-7 scores. First, both quarantine ($\beta = 0.062$, 95% CI [0.556–1.037], $p < 0.001$) and self-reported health ($\beta = 0.107$, 95% CI [0.921–1.321], $p < 0.001$) are significantly associated with GAD-7 scores, when both are included in the linear regression model. Second, in the formal interaction test between quarantine and self-reported health, the GAD-7 scores are statistically significant different among groups ($p < 0.001$). Finally, in the four groups, the results show that in the non-quarantined group, the relationship between GAD-7 scores and self-reported health is weak. Moreover, the difference in mean values of GAD-7 scores between the good and poor self-reported health groups is only 1.19. This difference increase to 1.34 in the quarantined group. Similarly, in the group of respondents with good self-reported health, the relationship between quarantine and GAD-7 scores is weak. Moreover, the difference in mean value of GAD-7 scores between the non-quarantined and quarantined groups is only 0.82. This difference increase to 0.97 in the group of respondents with poor self-reported health (Table 4). These results indicate that the interaction between quarantined and self-reported health status has a significant effect on anxiety (Fig. S2).

To evaluate whether there is a significant interaction effect between perceptions of COVID-19 and self-reported health on GAD-7 scores, the above analyses are duplicated using controlled/uncontrolled perceptions of COVID-19 as a dichotomous variable instead of quarantined/non-quarantined. The results suggest that perceptions of COVID-19 ($\beta = -0.076$, 95% CI [−1.198 to 0.729], $p < 0.001$) and self-reported health ($\beta = 0.107$, 95% CI [0.921–1.321], $p < 0.001$) are significantly associated with GAD-7 scores. Additionally, the interaction test between self-reported health and perceptions of COVID-19 yield a statistically significant result ($p < 0.001$). Multiple group comparisons

show that in the group of respondents with perceptions of COVID-19 being controlled, the relationship between GAD-7 scores and self-reported health is weak. Moreover, the difference in mean value of GAD-7 scores between the good and poor self-reported health groups is only 1.11. This difference increase to 1.57 in the group of respondents with perception that COVID-19 is uncontrolled. Likewise, in the group of respondents with good self-reported health, the relationship between perceptions of COVID-19 and GAD-7 scores is weak. Moreover, the difference in mean values of GAD-7 scores between the groups with perceptions of COVID-19 being controlled and uncontrolled is 0.92. This difference increased to 1.38 in the group with poor self-reported health (Table 4). These results suggest a significant effect on anxiety by the interaction between different levels of perceptions of COVID-19 and self-reported health status (Fig. S3).

## DISCUSSION

Overall, the prevalence of generalized anxiety was 37.69%, and female sex, age <40 years, living in rural areas, poor self-reported health, chronic diseases, and quarantine were risk factors for anxiety, while the perception that COVID-19 can be controlled was a protective factor. Additionally, ordinal logistic regression analysis reveal that self-reported health status, chronic diseases, quarantine, and perceptions of COVID-19 are significantly correlated with the severity of anxiety. Furthermore, the findings indicate that the interaction between quarantined and self-reported health status and between perceptions of COVID-19 and self-reported health status have a significant effect on generalized anxiety.

This study report a relatively high prevalence of anxiety among the participants, after the peak of the COVID-19 outbreak in China. This result may have been related to the closure of schools and enterprises, economic stagnation, and deteriorating social conditions, owing to which, the anxiety experienced by individuals is compounded (*Van Bortel et al., 2016*). In addition, it find that self-reported rates of anxiety are lower than those reported by front-line medical staff in China (*Zhou et al., 2020*) and higher than those reported by Chinese teachers (*Li et al., 2020*). This difference could be attributed to population segmentation. Specifically, compared with the general population, front-line medical staff with heavy workloads and stress are more likely to suffer from greater psychological distress (*Zhang et al., 2020*), whereas teachers, who comprise a highly educated group with relatively stable incomes, are less likely to experience anxiety (*Erik, 2020*). Therefore, future research needs to focus on different groups to understand the true burden of anxiety during an emergency public health crisis.

Previous studies have reported that generalized anxiety disorder is more common among female participants than their male counterparts (*Mirón et al., 2019*; *Sanad, 2019*; *Gao, Ping & Liu, 2020*), which is also confirmed in this study. In addition, this study report that during the COVID-19 outbreak, younger participants (<40 years) show significantly higher levels of anxiety than older participants (≥40 years), which is consistent with a previous study in Taiwan during the SARS outbreak (*Su et al., 2007*). Given that young people, who form part of the most productive segments of society, are more concerned about the future and economic impact of the COVID-19 pandemic and have greater access

to social media for information on high transmission and fatality rates, these factors may contribute to their anxiety (*Huang & Zhao, 2020*; *Ahmed et al., 2020*; *Moghanibashi-Mansourieh, 2020*). The factor of living in rural areas is an independent predictor of high anxiety in this study. This result is supported by evidence (*Zhou et al., 2020*) that the possible reasons for the rural-urban disparity may be related to differences in education levels and medical conditions (*Moore et al., 2005*; *Probst et al., 2002*).

This study found that poor self-reported health is a risk factor for anxiety. More importantly, the effects of self-reported health status on anxiety also interact with other factors. For example, the association of quarantine with anxiety is less in participants with good self-reported health than poor self-reported health. Similarly, the association of self-reported health with anxiety is lower in participants who are not quarantined than those who are quarantined. In other words, the most serious anxiety occurs in participants with poor self-reported health coupled with quarantine. This study also find that self-reported health and perceptions of COVID-19 have an interaction effect on anxiety. These significant interactions highlight the potentially harmful effects of poor self-reported health on participants' anxiety, especially if they are quarantined or have perceptions of COVID-19 being uncontrollable. We hope these findings will provide new insights into anxiety research and can be used to develop psychological interventions to improve the mental health of high-risk groups during epidemics.

This study show that participants with chronic disease have significantly higher levels of anxiety, which is consistent with earlier studies (*Bernstein, 2016*; *Renna et al., 2018*). People with chronic diseases are also likely to develop related disorders, such as anxiety, which, in turn, negatively impacts mental health and physical activities and ultimately leads to poor prognosis and disease control. Therefore, developing and applying appropriate mental health interventions for people with chronic diseases during the COVID-19 pandemic is essential.

During the outbreak of a disease, quarantine is a vital measure to protect public health (*Tognotti, 2013*; *Parmet & Sinha, 2020*). To date, limited studies have investigated the relationship between quarantine and anxiety during COVID-19. A study conducted during the outbreak of the Middle East respiratory syndrome indicated that those quarantined had higher levels of anxiety compared with healthy individuals (*Jeong et al., 2016*). Similar results are found in this study. Therefore, more supportive mental health care should be provided to those who have been quarantined during a pandemic.

Given that, in this study, perceiving that COVID-19 can be controlled is a significant protective factor for anxiety, positive psychological responses are associated with a lower prevalence of anxiety. In line with this result, a previous study reported that participants who were confident during the pandemic about COVID-19 being controllable were less likely to develop anxiety (*Zou et al., 2021*). These findings suggest that for reducing their anxiety levels, individuals should increase their knowledge about infectious diseases as well as their confidence in coping with the dilemma of public health emergencies.

This study has some limitations. First, its cross-sectional design prevents inferring any causal relationships between the variables and anxiety. Therefore, further studies should be conducted using longitudinal methods. Second, as face-to-face interactions were not

possible during the COVID-19 outbreak, levels of anxiety and physical symptoms were assessed using self-report scales, which may not be as accurate as an assessment by health professionals. More professional and precise methods, such as clinical diagnostic tools, should be used in future research. Third, this survey was conducted online; hence, those who had no Internet access were excluded, which may have contributed to selection bias.

## CONCLUSIONS

This study revealed that participants with the following demographics: female, young (<40 years), living in rural areas, poor self-reported health, chronic diseases, and quarantined were vulnerable to anxiety, while perceiving that COVID-19 could be controlled was a protective factor for anxiety. Moreover, the interactions between quarantine and self-reported health and between perceptions of COVID-19 and self-reported health were found to have a significant effect on anxiety. Taken together, these findings provide a new perspective on the study of generalized anxiety among the general population in China and may help to provide information for current and future mental health care services.

## ACKNOWLEDGEMENTS

The authors would like to thank the participants for their involvement in this study.

### Funding

This work was supported by the National Social Science Fund of China (20VYJ069) and Shenzhen social science planning project (SZ2020B019). The funders had no role in study design, data collection and analysis, decision to publish, or preparation of the manuscript.

### Grant Disclosures

The following grant information was disclosed by the authors:
National Social Science Fund of China: 20VYJ069.
Shenzhen Social Science Planning Project: SZ2020B019.

### Competing Interests

The authors declare that they have no competing interests.

### Author Contributions

- Yi Xia conceived and designed the experiments, performed the experiments, analyzed the data, prepared figures and/or tables, authored or reviewed drafts of the article, and approved the final draft.
- Qi Wang conceived and designed the experiments, performed the experiments, authored or reviewed drafts of the article, and approved the final draft.
- Lushaobo Shi performed the experiments, prepared figures and/or tables, authored or reviewed drafts of the article, and approved the final draft.
- Zengping Shi performed the experiments, prepared figures and/or tables, and approved the final draft.

- Jinghui Chang analyzed the data, authored or reviewed drafts of the article, and approved the final draft.
- Richard Xu analyzed the data, authored or reviewed drafts of the article, and approved the final draft.
- Huazhang Miao performed the experiments, analyzed the data, authored or reviewed drafts of the article, and approved the final draft.
- Dong Wang conceived and designed the experiments, performed the experiments, authored or reviewed drafts of the article, and approved the final draft.

## Ethics

The following information was supplied relating to ethical approvals (*i.e.*, approving body and any reference numbers):

The study was conducted with the approval of the Ethics Committee of Southern Medical University (Ref ID: NFYKDX002).

## Data Availability

The raw measurements are available in the Supplemental Files.

## Supplemental Information

Supplemental information for this article can be found online at http://dx.doi.org/10.7717/peerj.14720#supplemental-information.

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
