# Peer review of "Prevalence and risk factors of COVID-19-related generalized anxiety disorder among the general public in China: a cross-sectional study"

_PeerJ, doi:10.7717/peerj.14720_

## Round 0.1 · original submission · Major Revisions

Dear Authors,

Your paper needs major revisions. I invite you to adapt your manuscript and answer in detail the comments of reviewer 2 and reviewer 3 about the methodology and the interpretation of the results.

Reviewer 1 has requested that you cite specific references. I do not expect you to include these citations.

Reviewer 1 ·

Basic reporting

I have the following comment for the authors to address.

1) The authors stated "Currently COVID-19 has overwhelmed the health
69 care system and imposed a heavy social and economic burden on individuals, families,
70 communities and countries.3-" I recommend the authors to measure the impact on mental health based on the following studies:

A systematic review of COVID-19 on mental health
Impact of COVID-19 pandemic on mental health in the general population: A systematic review [published online ahead of print, 2020 Aug 8]. J Affect Disord. 2020;277:55-64. doi:10.1016/j.jad.2020.08.001

The impact of COVID-19 on three continents and its relationship with physical health:
A chain mediation model on COVID-19 symptoms and mental health outcomes in Americans, Asians and Europeans. Sci Rep 11, 6481 (2021). https://doi.org/10.1038/s41598-021-85943-7

The impact of COVID-19 on developing countries:
The impact of COVID-19 pandemic on physical and mental health of Asians: A study of seven middle-income countries in Asia. PLoS One. 2021 Feb 11;16(2):e0246824. doi: 10.1371/journal.pone.0246824. PMID: 33571297.

Government response during the pandemic:
Government response moderates the mental health impact of COVID-19: A systematic review and meta-analysis of depression outcomes across countries. J Affect Disord. 2021 May 27;290:364-377. doi: 10.1016/j.jad.2021.04.050. Epub ahead of print. PMID: 34052584.

Worst outcome of COVID infection due to depression

Association Between Mood Disorders and Risk of COVID-19 Infection, Hospitalization, and Death: A Systematic Review and Meta-analysis. JAMA Psychiatry. 2021 Jul 28. doi: 10.1001/jamapsychiatry.2021.1818. Epub ahead of print. PMID: 34319365.

Post COVID and depression

Onset and frequency of depression in post-COVID-19 syndrome: A systematic review. J Psychiatr Res. 2021 Dec;144:129-137. doi: 10.1016/j.jpsychires.2021.09.054. Epub 2021 Sep 30. PMID: 34619491; PMCID: PMC8482840.

Cognitive impairment

Fatigue and Cognitive Impairment in Post-COVID-19 Syndrome: A Systematic Review and Meta-Analysis. Brain Behav Immun. 2021 Dec 29:S0889-1591(21)00651-6. doi: 10.1016/j.bbi.2021.12.020. Epub ahead of print. PMID: 34973396.

2) The authors stated " Second, drastic lockdown measures, including quarantine, home isolation, restricting 75 gatherings, and prohibition of many human activities may lead to unemployment and economic 76 downturn, subsequently leading to anxiety7 8". Please include the impact of face mask and social distancing.

The Association Between Physical and Mental Health and Face Mask Use During the COVID-19 Pandemic: A Comparison of Two Countries With Different Views and Practices. Front Psychiatry. 2020;11:569981Published 2020 Sep 9. doi:10.3389/fpsyt.2020.569981
.
Impact of COVID-19 on Economic Well-Being and Quality of Life of the Vietnamese During the National Social Distancing. Front Psychol. 2020 Sep 11;11:565153. doi: 10.3389/fpsyg.2020.565153. PMID: 33041928; PMCID: PMC7518066.

3) The authors stated "Recently, some studies have focused on assessing the degree of anxiety disorders in specific 82 populations". Please include the impact on adolescents and workers:

Impact on Adolescents

What Factors Are Most Closely Associated With Mood Disorders in Adolescents During the COVID-19 Pandemic? A Cross-Sectional Study Based on 1,771 Adolescents in Shandong Province, China. Front Psychiatry. 2021 Sep 16;12:728278. doi: 10.3389/fpsyt.2021.728278. PMID: 34603106; PMCID: PMC8481827.

Impact on workers
Is Returning to Work during the COVID-19 Pandemic Stressful? A Study on Immediate Mental Health Status and Psychoneuroimmunity Prevention Measures of Chinese Workforce [published online ahead of print, 2020 Apr 23]. Brain Behav Immun. 2020;S0889-1591(20)30603-6. doi:10.1016/j.bbi.2020.04.055

4) Under the discussion, the authors stated "Our study reported a relatively high prevalence of anxiety among the participants after the peak
PeerJ reviewing PDF | (2022:03:71827:0:1:NEW 29 Mar 2022)
Manuscript to be reviewed", can you compare the prevalence with the following studies at the peak in China?

Immediate Psychological Responses and Associated Factors during the Initial Stage of the 2019 Coronavirus Disease (COVID-19) Epidemic among the General Population in China. Int J Environ Res Public Health. 2020;17(5):1729. Published 2020 Mar 6. doi:10.3390/ijerph17051729

A Longitudinal Study on the Mental Health of General Population during the COVID-19 Epidemic in China [published online ahead of print, 2020 Apr 13]. Brain Behav Immun. 2020; S0889-1591(20)30511-0. doi:10.1016/j.bbi.2020.04.028

5) The authors stated ". People with chronic diseases are likely to develop
289 related disorders such as anxiety" Please provide an example based on lupus:

. Damage accrual, cumulative glucocorticoid dose and depression predict anxiety in patients with systemic lupus erythematosus. Clin Rheumatol. 2011 Jun;30(6):795-803. doi: 10.1007/s10067-010-1651-8. Epub 2011 Jan 11. PMID: 21221690.

6) Under the discussion, the authors should discuss how to use online psychological intervention to help people in China during COVID-19 pandemic. Please refer to the following studies:

The most evidence-based treatment is cognitive behaviour therapy (CBT), especially Internet CBT that can prevent the spread of infection during the pandemic.

Use of Cognitive Behavior Therapy (CBT) to treat psychiatric symptoms during COVID-19:
Mental Health Strategies to Combat the Psychological Impact of COVID-19 Beyond Paranoia and Panic. Ann Acad Med Singapore. 2020;49(3):155‐160.

Cost-effectiveness of iCBT:
Moodle: The cost effective solution for internet cognitive behavioral therapy (I-CBT) interventions. Technol Health Care. 2017;25(1):163-165. doi: 10.3233/THC-161261. PMID: 27689560.

Internet CBT can treat psychiatric symptoms such as insomnia:
Efficacy of digital cognitive behavioural therapy for insomnia: a meta-analysis of randomised controlled trials. Sleep Med. 2020 Aug 26;75:315-325. doi: 10.1016/j.sleep.2020.08.020. Epub ahead of print. PMID: 32950013.

7) The authors stated "e. Therefore, self-report scales were used to assess levels of anxiety and physical
311 symptoms, which may not be as accurate as the assessment by health professionals". Please mention the gold standard in psychological assessment:

The COVID-19 pandemic was found to cause hemodynamic changes in the brain (Olszewska-Guizzo et al 2021) and impairment in olfactory function (Ho et al 2021). This study mainly used self-reported questionnaires to measure psychiatric symptoms and did not make clinical diagnosis. The gold standard for establishing psychiatric diagnosis involved structured clinical interview and functional neuroimaging should be applied in the future face-to-face research after COVID-19 restrictions are removed. (Husain et al 2019, Husain et al 2020, Ho et al 2020).

References:

Olszewska-Guizzo, A.; Mukoyama, A.; Naganawa, S.; Dan, I.; Husain, S.F.; Ho, C.S.; Ho, R. Hemodynamic Response to Three Types of Urban Spaces before and after Lockdown during the COVID-19 Pandemic. Int. J. Environ. Res. Public Health 2021, 18, 6118. https://doi.org/10.3390/ijerph18116118

Ho RC et al Comparison of Brain Activation Patterns during Olfactory Stimuli between Recovered COVID-19 Patients and Healthy Controls: A Functional Near-Infrared Spectroscopy (fNIRS) Study. Brain Sciences. 2021; 11(8):968. https://doi.org/10.3390/brainsci11080968

Husain SF, Yu R, Tang TB, et al. Validating a functional near-infrared spectroscopy diagnostic paradigm for Major Depressive Disorder. Sci Rep. 2020;10(1):9740. Published 2020 Jun 16. doi:10.1038/s41598-020-66784-2

Husain SF, Tang TB, Yu R,et al . Cortical haemodynamic response measured by functional near infrared spectroscopy during a verbal fluency task in patients with major depression and borderline personality disorder. EBioMedicine. 2019 Dec 23;51:102586. doi: 10.1016/j.ebiom.2019.11.047. PMID: 31877417.

Ho CSH, Lim LJH, Lim AQ, et al. Diagnostic and Predictive Applications of Functional Near-Infrared Spectroscopy for Major Depressive Disorder: A Systematic Review. Front Psychiatry. 2020;11:378. Published 2020 May 6. doi:10.3389/fpsyt.2020.00378

Experimental design

The design is fine.

Validity of the findings

The findings are valid

Reviewer 2 ·

Basic reporting

no comment

Experimental design

1)The abstract said stratified sampling was carried out, but the method of sampling was not reported. The questionnaire distribution, issuing institutions and the way to expand the sample size were not reported.
2)No sample size estimation was performed.
3) Potential confounding factors were not described.
4) There is no description of the determination method of important factors. For example, What are the criteria for judging Perception of COVID-19?Just two options are not enough to get the right information.
5)The result data should not appear in the method section, line 108.
6)It is unreasonable to exclude questionnaires that are longer than 20 minutes.
7) The article does not report how missing values are handled.
8) The article does not report methods of logical detection between problems.
9) The method of interaction detection is not suitable.

Validity of the findings

1) The representation of the sample cannot be judged without reporting the number of people in each province.
2) The article did not report response rates. It is recommended to use a flow chart to show the variation in the number of participants.
3) Confounding factors are not explored and discussed. There are too few factors involved and the results may not be robust.
4) It is too simple to have only two answers to all the major questions.
5) Why does the paper only analyze the interaction of the two factors?
6) line 244-247, What are the principles for selecting protective factors? Why not say male is a protective factor? Why not think uncontrollable is a risk factor?
7) The representativeness of the study sample is unknown, and prevalence is not reliable.
8) line 252-253, This result is not supported by the literature. Why is the prevalence high in China?
9) Self-reported health results are too vague with only two choices.

Reviewer 3 ·

Basic reporting

Basic reporting is adequate.

Experimental design

Method section and Discussion section is correct. However, reporting needs improvements.

Validity of the findings

Findings are valid.

Additional comments

Thanks for opportunity to review manuscript entitled ' 'Prevalence and risk factors of COVID-19-related generalized anxiety disorder among the general public in China: A cross-sectional study'' for Peerj journal. The article is well-written. However, some minor correction required before publication of article.
1. Showing statistical symbols are wrong along to manuscript. Authors must correct this problem as per APA 7 rules.
2. Interaction best understand with figures. Authors must add figures for their interaction analyses.
3. Practical implications are completely missing in the manuscript and must be added.
4. Semi-partial correlations must be added to Table 3.
5. Abstract section must rearrange using Background, Method, Results, and Conclusion Section.
6. Authors must give more information about ethical aspects of their study such as anonymity and voluntary participation.
7. As a multivariate analyst, I think both using linear regression and ordinal logistic regression is redundant. Authors solely focus on linear regression. Testing robustness of findings with ordinal logistic regression is unnecessary. If authors want to test robustness of findings, they can conduct bootstrap regression analyses. However, their ordinal findings is interesting. If authors want to keep this analyses, they must give convincing reasons other than robustness.

---

## Round 0.2 · Major Revisions

The description of the methodology is still too vague. Please address the comments from reviewers to improve the description and understanding of the study

Reviewer 2 ·

Basic reporting

No comment.

Experimental design

1. China was divided into three sub-regions (eastern, central, and western China) according to the geographical and economic development level. I think that means you think there might be potential differences in the results in the three regions, But you didn't compare the results by regions.
2. China was divided into three sub-regions (eastern, central, and western China) according to the geographical and economic development level. Are there any references to prove that the division is based on economic level?
3. You said that you had added the description of the criteria for judging Perception of COVID-19, line 131. But, I didn't find them.
4. Were the 27 articles excluded for logical reasons all because of consistent choices? I don't think logical exclusion should be that simple.
5. The question of confounding factors is too simple. Vaccine issues, isolation factors are all common factors of COVID-19 that are not analyzed, especially when it comes to mental health.
6. Your question is only two choices, is it to prove that one is a protective factor and the other is a risk factor? How did you choose the wording of risk factors and protective factors? You say women are a risk factor, why not men are a protective factor?
7. I still don't think it makes sense to have only two options when considering these outcomes, because many people's health experience is not an extreme condition. Even if you say it's because of the convenience of the respondents and the increased response rate.

Validity of the findings

The suggestion was made in the previous section.

Reviewer 3 ·

Basic reporting

Basic reporting is adequate.

Experimental design

Method and result section contain minor problems.

Validity of the findings

The findings are valid.

Additional comments

Thanks for opportunity review revised manuscript entitled ' ' Prevalence and risk factors of COVID-19-related generalized anxiety disorder among the general public in China: A cross-sectional study'' for Peerj journal. Three minor revisions required before publication of article.
1. All p values must be small and italic along the manuscript.
2. 95% CI must not be italic along the the manuscript.
3. In the Multiple regression analysis of factors associated with anxiety during COVID-19 section of manuscript all following statements (OR = 1.184, 95% CI: 1.078, 1.301, P < .001) must remove, It is mind confusing and impossible to calculate odd ratio in multiple linear regression analysis.

---

## Round 0.3 · Minor Revisions

Please correct the revision requested by reviewer 1 and resubmit the paper for another review. Thank you

Reviewer 2 ·

Basic reporting

no comment

Experimental design

1. As I suggested last time, you added specific analysis results are shown in table 1 to 9. These tables are too jumbled; they should be consolidated into large tables. Then use it as part of a formal essay or as supplementary material.
2. You used WJX.cn to create a questionnaire, so multiple choice is not going to happen in single-choice questions. So, your reply does not solve my question about the deletion logic exclusion. I wonder if you set up any inconsistent questions to judge logic, as opposed to the simple questions of the questionnaire itself.
3. The question about risk factor, you said that in terms of gender (OR = 1.200, p < .001), and the reference is male, so we say female is a risk factor. Why the reference is male?
4. You said that you will set up multiple consecutive items in the follow-up research. I look forward to your next research.

Validity of the findings

The suggestion was made in the previous section.

Reviewer 3 ·

Basic reporting

Basic reporting is correct and adequate.

Experimental design

Method and results section is correct.

Validity of the findings

The findings are valid.

Additional comments

Thanks for opportunity review revised manuscript entitled "Prevalence and risk factors of COVID-19-related generalized anxiety disorder among the general public in China: A cross-sectional study'' for Peerj journal.Authors revised the manuscript with a good will and now it is correct. I recommend accept decision.

---

## Round 0.4 · Minor Revisions

We received a last review of your document. Please address the different comments and resubmit the paper.
Best

Reviewer 4 ·

Basic reporting

The presented article is interesting and provides relevant research. As the article has already been improved according to previous reviews, there are some minor suggestions for improvement of the manuscript.
I suggest that the article is proofread by English professional or fluent English speaker.

Experimental design

Research design is relevant for the situation at the beginning of pandemic. Methods are well described, some minor corrections are suggested to improve the text

Validity of the findings

All underlying data have been provided; they are robust and statistically sound , however the explanation of data in text should be improved.
Conclusions are well stated, linked to original research question & limited to supporting results.

Additional comments

Some suggestions to improve the manuscript and make content more understandable to the reader
- In Materials and methods chapter: Interaction effect, lines 132-145. If I understand methodology correctly, it would be better to state: e.g. respondents were classified in four categories and GAD-7 score values were compared among groups…
- In Results chapter:
o use present simple tense when you refer to presentation of your results
 Line 160: number of participants was shown
 Line 161: The number of people in each province were listed in Supplemental Table 1….
o Lines 166, 167: if p>0,05, then it is not relevant to list demographic factor in terms of majority.
o Lines 169-173: it is not clear what are p-values indicating. Comparison with some expected values? Differences among regions?
o If p-values in lines 166-173 are referring to unequal distribution of GAD7 among groups, this should be clearly explained.
o Line 199: instead 'the GAD-7 scores obtained a statistically significant result' would be probably better to say, 'the GAD-7 scores were statistically significant different among groups '?
o Line 208: 'Supplementary figure 2' should be put in brackets or stated '… as shown in Supplementary figure 2'. Same in line 225.
- I suggest that the article is proofread by English professional or fluent English speaker.
o E.g. line 219-220:' This difference increased to 1.57 in the perceptions of COVID-19 as uncontrolled group.' – probably better: ' This difference increased to 1.57 in the group of respondents with perception that COVID-19 is uncontrolled.'

---

## Round 0.5 · accepted · Accept

Dear Authors

Thank you for the revisions of your manuscript. You have answered all requests from the reviewers and I am satisfied with your answers.